# A pH- and ionic strength-dependent conformational change in the neck region regulates DNGR-1 function in dendritic cells

Pavel Hanč[1], Oliver Schulz[1], Hanna Fischbach[1], Stephen R Martin[2], Svend Kjær[2] & Caetano Reis e Sousa[1,*]

## Abstract

DNGR-1 is receptor expressed by certain dendritic cell (DC) subsets and by DC precursors in mouse. It possesses a C-type lectin-like domain (CTLD) followed by a poorly characterized neck region coupled to a transmembrane region and short intracellular tail. The CTLD of DNGR-1 binds F-actin exposed by dead cell corpses and causes the receptor to signal and potentiate cross-presentation of dead cell-associated antigens by DCs. Here, we describe a conformational change that occurs in the neck region of DNGR-1 in a pH- and ionic strength-dependent manner and that controls cross-presentation of dead cell-associated antigens. We identify residues in the neck region that, when mutated, lock DNGR-1 in one of the two conformational states to potentiate cross-presentation. In contrast, we show that chimeric proteins in which the neck region of DNGR-1 is replaced by that of unrelated C-type lectin receptors fail to promote cross-presentation. Our results suggest that the neck region of DNGR-1 is an integral receptor component that senses receptor progression through the endocytic pathway and has evolved to maximize extraction of antigens from cell corpses, coupling DNGR-1 function to its cellular localization.

**Keywords** C-type lectin receptors; cross-presentation; necrosis; protein structure

**Subject Categories** Immunology

**The EMBO Journal (2016) 35: 2484–2497**

## Introduction

Recognition of damage-associated molecular patterns (DAMPs), molecules released or exposed by cells upon injury, is essential for maintenance of homeostasis and tissue repair (Zelenay & Reis e Sousa, 2013). In vertebrates, it can also lead to adaptive immune responses against proteins present within dead cells, including cancer neoantigens (Zelenay & Reis e Sousa, 2013). DNGR-1 (also known as CLEC9A) is a vertebrate DAMP receptor expressed by certain types of dendritic cells (DCs), leukocytes that couple innate and adaptive immunity. DNGR-1 specifically binds to a DAMP exposed by cells that have lost their plasma membrane integrity (Sancho *et al*, 2009). We and others have previously identified the DNGR-1 ligand as the filamentous form of actin (F-actin) (Ahrens *et al*, 2012; Zhang *et al*, 2012) and determined the structure of the ligand–receptor complex (Hanč *et al*, 2015). DNGR-1 can act as an endocytic receptor, yet it is dispensable for uptake of dead cell debris by DCs (Huysamen *et al*, 2008; Sancho *et al*, 2008, 2009). Rather, DNGR-1 recognition of F-actin favors the cross-presentation of dead cell-associated antigens by DCs that internalized dead cell debris (Sancho *et al*, 2009; Iborra *et al*, 2012; Zelenay *et al*, 2012). The exact mechanism by which DNGR-1 exerts this function remains elusive but is thought to involve regulation of the maturation of endosomes containing dead cell material (Sancho *et al*, 2009; Iborra *et al*, 2012; Zelenay *et al*, 2012).

DNGR-1 is a type II transmembrane protein and a member of the C-type lectin superfamily. In mouse, multiple DNGR-1 isoforms have been found, of which only two retain the entire ligand-binding domain and the transmembrane region (termed "*long*" and "*short*") (Huysamen *et al*, 2008; Sancho *et al*, 2008). In human, a single isoform exists that corresponds to the mouse "*short*" receptor and behaves as a glycosylated homodimer (Huysamen *et al*, 2008). In contrast, the mouse receptor has been described as non-glycosylated and monomeric in one report (Huysamen *et al*, 2008) or as a glycosylated dimer in another (Sancho *et al*, 2008). The extracellular domain of DNGR-1 consists of a single C-type lectin-like domain (CTLD) and a membrane-proximal neck region. The neck of the *long* isoform of murine DNGR-1 differs from the *short* and the human isoforms by the presence of an extra exon coding for additional 26 amino acids (Huysamen *et al*, 2008; Sancho *et al*, 2008). The intracellular portion of DNGR-1 contains a short domain termed hemITAM, which resembles that found in the related C-type lectin

1   Immunobiology Laboratory, The Francis Crick Institute, London, UK
2   Structural Biology Science Technology Platform, The Francis Crick Institute, London, UK
    *Corresponding author. Tel: +44 20 3796 1310; E-mail: caetano@crick.ac.uk

dectin-1 (Rogers *et al*, 2005) and which permits signaling via Syk (Huysamen *et al*, 2008; Sancho *et al*, 2009).

The CTLD of human DNGR-1 has been crystalized (Zhang *et al*, 2012) and shown to be solely responsible for the ability of DNGR-1 to bind to F-actin (Ahrens *et al*, 2012; Zhang *et al*, 2012; Hanč *et al*, 2015). On the other hand, the structure and properties of the neck region and their relevance to the biology of the receptor have not been explored. Although originally seen primarily as serving a scaffold function, the neck region is emerging as an important determinant of the properties of various C-type lectin receptors (Back *et al*, 2009; Tabarani *et al*, 2009; Manzo *et al*, 2012). Here, we report that the neck region of mouse and human DNGR-1 allows the receptor to exist in two distinct conformations that interconvert in response to changes in pH and ionic strength. The ability of DNGR-1 to adopt different conformations as a function of pH and ionic strength suggests that this receptor may be intrinsically altered as it travels through the endocytic pathway and we present data indicating that this contributes to its function in cross-presentation of dead cell-associated antigens.

## Results

### DNGR-1 is a glycosylated disulfide-bonded homodimer

To assess their properties, we expressed the entire extracellular domains (ECDs) of mouse DNGR-1 *long* and *short* isoforms, as well as the single human isoform, as soluble recombinant FLAG-tagged proteins (Ahrens *et al*, 2012; Hanč *et al*, 2015). By non-reducing SDS–PAGE followed by Western blotting for FLAG, all ECDs ran as dimers, with a minor fraction in the form of higher order oligomers (Fig 1A). The dimers and oligomers could be reduced to their constituent monomers using dithiothreitol (DTT) or β-mercaptoethanol (β-ME) (Fig 1A). Mutating the conserved cysteine in the neck region to serine (C94S in the *long* mouse isoform ECD) had a strong adverse effect on expression efficiency (data not shown) but resulted in a protein that ran as a monomer under both reducing and non-reducing conditions (Fig 1B). Finally, all ECDs were glycosylated as their electrophoretic mobility could be augmented, at least in part, by treatment with the glycosidase PNGase F (Fig 1C). The deglycosylation treatment did not change the two-band pattern of DNGR-1 ECDs mobility, consistent with a previous report (Huysamen *et al*, 2008). Importantly, PNGase F treatment only removes N-bound glycans which do not contain an α(1-3)-fucose bound to the core N-acetyl glucosamine. It is therefore possible that the two bands correspond to glycoforms that differ in O-glycosylation or α(1-3)-fucosylated N-glycans. In summary, the ECDs of both human and mouse DNGR-1 behave as glycosylated dimers.

### Low pH and ionic strength can induce formation of reduction-resistant dimers of DNGR-1

Surprisingly, we noticed that under certain conditions, the dimers of DNGR-1 could maintain their dimeric status even in the presence of reducing agents (Fig 2A). Buffers of lower pH and ionic strength were most effective in inducing the reduction-insensitive state (Fig 2A). We confirmed this by testing 36 buffers of different ionic strengths (range 15–250 mM) and pH (range pH 6.5–8.05) in which

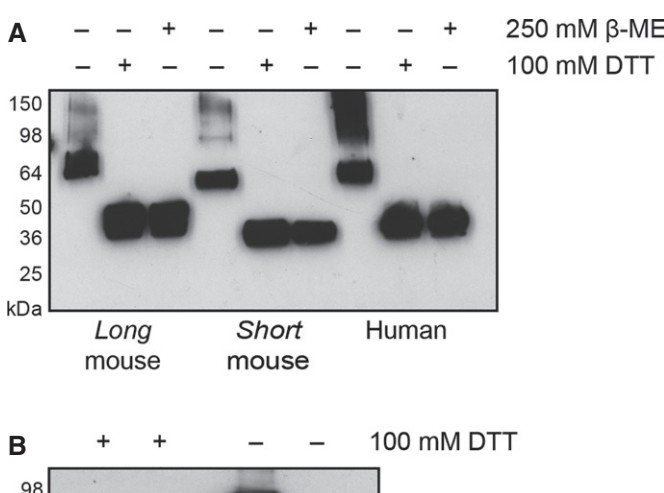

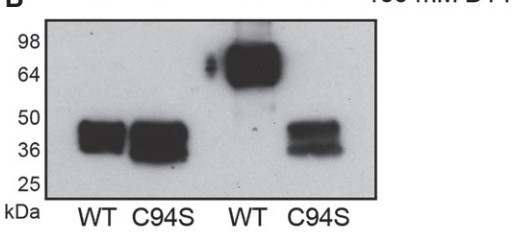

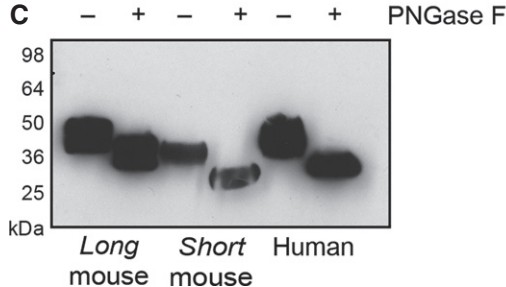

**Figure 1.  DNGR-1 is a glycosylated disulfide-bonded dimer.**

A  Western blot analysis of supernatants containing *long* and *short* mouse and human DNGR-1 ECD proteins under reducing and non-reducing conditions.

B  Western blot analysis of supernatants after protein production of WT and C94S mutant *long* mouse DNGR-1 ECD proteins under reducing and non-reducing conditions. C94S mutant-containing supernatant was 200× concentrated before analysis.

C  Western blot analysis of supernatants containing the indicated DNGR-1 ECD proteins after treatment with PNGase F overnight. HRP-conjugated anti-FLAG antibody was used for detection of all proteins. Numbers on the side of blots indicate positions of molecular weight markers.

Source data are available online for this figure.

buffering capacity was kept constant (Table 1). After analysis by SDS–PAGE in the presence of 100 mM DTT, we observed a clear trend toward higher abundance of reduction-insensitive dimers as a function of a decrease in pH and ionic strength (Fig 2B).

### A reversible conformational change is responsible for the formation of reduction-insensitive dimers

As the neck region is responsible for dimerization of DNGR-1, we hypothesized that a pH- and ionic strength-dependent

conformational change in this portion of the protein could be responsible for the reduction insensitivity of DNGR-1 dimers. To test this possibility, we subjected the reduction-insensitive form of *long* mouse DNGR-1 ECD to mildly denaturing (Laemmli buffer) or strongly denaturing (8 M urea) conditions in the presence of DTT and analyzed the samples by SDS–PAGE and Western blot. As predicted, strongly denaturing conditions almost completely abolished the ability of DNGR-1 to resist reduction, while under weakly denaturing conditions the reduction-insensitive dimers could be observed. In the absence of reducing agents, the protein maintained its dimeric status regardless of the denaturing conditions (Fig 2C). To the same end, we gradually increased the stringency of the heat denaturation step by increasing the time of boiling in Laemmli buffer. When analyzed by SDS–PAGE and

Western blot, DNGR-1 ECD appeared exclusively as a monomer after 5 min of boiling under neutral conditions, while the protein kept under mildly acidic conditions (MES pH 6.1) was not completely reduced even after 15 min of the same treatment (Fig 2D).

To evaluate the reversibility of the conformational change, we subjected *long* mouse DNGR-1 ECD to mildly acidic conditions (MES pH 6.1) to induce the reduction-resistant state, dialyzed the protein back into PBS, and tested its reduction sensitivity. As expected, we observed reduction-resistant dimers under mildly acidic conditions (Fig 3A). However, the protein that had been MES-treated and transferred back into PBS showed no sign of increased resistance to DTT treatment (Fig 3A), suggesting that the conformational change is indeed reversible.

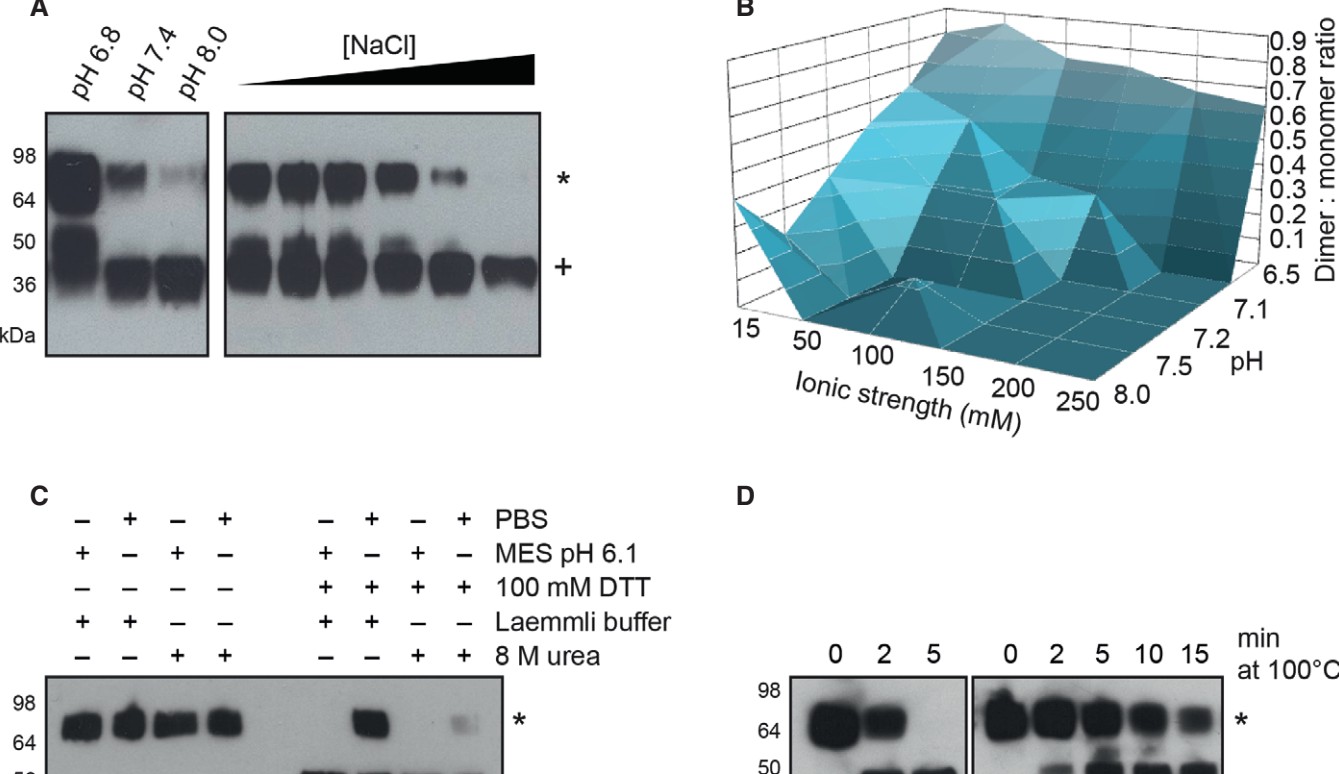

**Figure 2.  DNGR-1 can form reduction-resistant dimers.**

A  0.25 µg of purified *long* mouse DNGR-1 ECD was diluted into 10 mM Tris buffer of the indicated pH (left panel) or into 10 mM Tris pH 7.4 buffer supplemented with increasing amounts of NaCl (1–250 mM; right panel), and reduction sensitivity of DNGR-1 was assessed by reducing SDS–PAGE and Western blot.

B  0.25 µg of purified DNGR-1 *long* mouse ECD was diluted into buffers of different pH and ionic strength, and its reduction sensitivity was assessed by reducing SDS–PAGE and Western blot. The intensity of bands corresponding to dimer and monomer was determined densitometrically, and the ratio was plotted as a function of buffer ionic strength and pH.

C  0.25 µg of purified *long* mouse DNGR-1 ECD was diluted into PBS or 10 mM MES pH 6.1 buffer, and its reduction sensitivity was tested under mildly (Laemmli buffer) or strongly denaturing (8 M urea) conditions by reducing SDS–PAGE and Western blot.

D  0.25 µg of purified *long* mouse DNGR-1 ECD was diluted into PBS or 10 mM MES pH 6.1 buffers, and its reduction sensitivity after different lengths of heat denaturation in Laemmli buffer was tested by reducing SDS–PAGE and Western blot.

Data information: HRP-conjugated anti-FLAG antibody was used for detection of all proteins. Numbers on the side of blots indicate positions of molecular weight markers. Reduction-resistant dimers are indicated with an * and reduction-sensitive protein with a +.

Source data are available online for this figure.

**Table 1.  Buffers used to assess the influence of pH and ionic strength on the conformational state of DNGR-1.**

| Buffer | pH | Ionic strength (mM) | | | | | |
|---|---|---|---|---|---|---|---|
| 15 mM Bis-Tris | 6.5 | 15 | 50 | 100 | 150 | 200 | 250 |
| 15 mM BES | 7.1 | 15 | 50 | 100 | 150 | 200 | 250 |
| 15 mM MOPS | 7.2 | 15 | 50 | 100 | 150 | 200 | 250 |
| 15 mM HEPES | 7.5 | 15 | 50 | 100 | 150 | 200 | 250 |
| 15 mM Tricine | 8.05 | 15 | 50 | 100 | 150 | 200 | 250 |
| 15 mM Trizma base | 8.06 | 15 | 50 | 100 | 150 | 200 | 250 |

Finally, to confirm that the ability of the ECD to undergo a conformational change is not an isoform-specific property, we tested the reduction sensitivity of the two mouse isoforms and the human isoform. Under neutral conditions (PBS), all the proteins were readily reducible while, when subjected to weakly acidic conditions (pH 6.1), they showed increased resistance to reduction (Fig 3B). These data indicate that the ability to undergo a pH-dependent conformational change is an evolutionarily conserved property of DNGR-1.

### DNGR-1 neck region is necessary and sufficient for the formation of reduction-resistant dimers

We expressed a chimeric ECD, in which the neck region of DNGR-1 was fused to the CTLD of another C-type lectin receptor, dectin-1 (Fig 4A). Dectin-1 is a monomer but the chimeric ECD behaved as a disulfide-bonded homodimer (Fig 4B), consistent with the fact that the neck region of DNGR-1 bears the requisite cysteine for receptor dimerization (see above). Importantly, the chimeric protein was

refractory to DTT-mediated reduction under the same conditions that induced the reduction-insensitive state in DNGR-1 ECD (Fig 4C). Dectin-1 alone showed no such phenotype (Fig 4C). We conclude that the neck region of DNGR-1 is necessary and sufficient for the formation of reduction-resistant dimers.

### The conformational change happens at the level of tertiary structure

In order to further characterize the conformational change, we made use of circular dichroism (CD). Far-UV CD spectra reflect the secondary structure content of the protein while near-UV CD signals derive from the three aromatic residues (Phe, Tyr, and Trp) and, in some cases, from disulfide bonds, and reflect protein tertiary structure (Martin & Schilstra, 2008). We first measured far-UV CD spectra for both *long* mouse and human DNGR-1 isoforms under neutral (PBS) or mildly acidic (MES pH 6.1) conditions, and observed almost perfect overlap between the two (Fig 5A). Detailed analysis of both mouse and human isoforms revealed only a minor increase in the content of alpha helical structure under low pH conditions, corresponding at most to four residues (Table 2). Interestingly, a different content of disordered structure was observed in the mouse versus human isoform (44% versus 37%) corresponding to ~26 residues. This is precisely the length of the stretch of amino acids encoded in the extra exon present in the *long* mouse isoform, suggesting that this part of the mouse protein is unstructured.

With no major change in the secondary structure, we envisaged that mutual repositioning of the two neck regions within the dimer might be at the root of the observed reduction resistance. To test this hypothesis, we measured near-UV CD spectra of *long* mouse

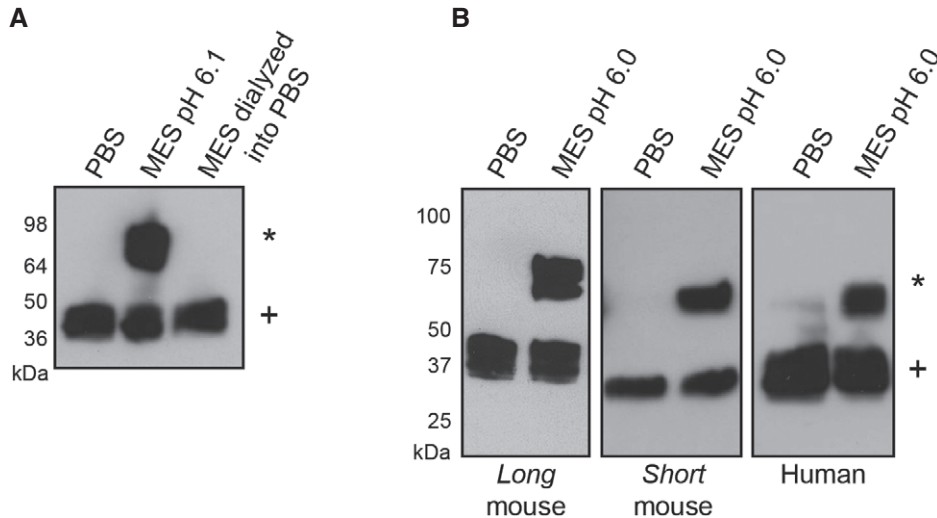

**Figure 3.  Formation of reduction-resistant dimers is reversible and conserved between DNGR-1 isoforms.**

A    1 μg of purified *long* mouse DNGR-1 ECD was transferred into 10 mM MES pH 6.1 and 0.5 μg into PBS. Half of the MES sample and the whole PBS sample were dialyzed against 2 l of PBS overnight at 4°C. After dialysis, all samples were prepared for reducing SDS–PAGE and Western blot.

B    Supernatants after production of indicated murine proteins in 293F cells were harvested and diluted into indicated buffers at a 2:3 ratio. 1 μg of purified human DNGR-1 ECD was diluted into the same buffers and all samples were analyzed using reducing SDS-PAGE and Western blot.

Data information: HRP-conjugated anti-FLAG antibody was used for detection of all proteins. Numbers on the side of blots indicate positions of molecular weight markers. Reduction-resistant dimers are indicated with an * and reduction-sensitive protein with a +.
Source data are available online for this figure.

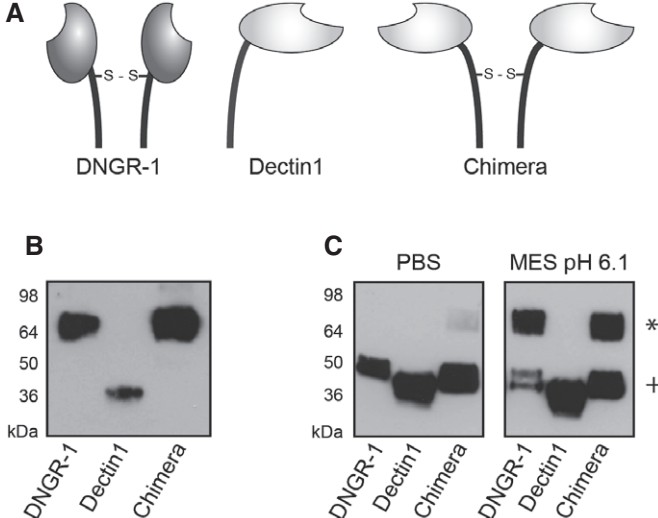

**Figure 4. The neck of DNGR-1 is sufficient for formation of reduction-resistant dimers.**

A  Schematic representation of DNGR-1, dectin-1, and the chimeric protein consisting of the neck region of DNGR-1 and CTLD of dectin-1.

B  Supernatants containing ECD of *long* mouse DNGR-1, dectin-1, and the chimeric receptor were prepared for SDS–PAGE and Western blot under non-reducing conditions.

C  Supernatants containing indicated proteins were diluted into PBS or pH 6.1 MES buffer and analyzed using SDS–PAGE and Western blot under reducing conditions.

Data information: HRP-conjugated anti-FLAG antibody was used for detection of all proteins. Numbers on the side of blots indicate positions of molecular weight markers. Reduction-resistant dimers are indicated with an * and reduction-sensitive protein with a +.

Source data are available online for this figure.

DNGR-1 under neutral and mildly acidic (pH 6.1) conditions (Fig 5B). The sample in PBS showed significant near-UV CD intensity extending to long wavelengths (> 350 nm), characteristic of a contribution from a disulfide bond. This long wavelength intensity was not observed in the low pH sample, suggesting that the disulfide dihedral angle is different at the two pHs. Although the difference spectrum (PBS–MES) is characteristic of a disulfide contribution it is also possible that the changes observed around 290 nm indicate that there is a change in the environment of at least one tryptophan residue. Thus, taken together, our near-UV and far-UV CD data suggest that a change in the tertiary but not secondary structure happens when DNGR-1 undergoes the transition to the reduction-insensitive form. Our data are consistent with a model in which repositioning of the neck regions within the dimer results in protection of the disulfide bond or in making the disulfide bond dispensable for maintaining the dimeric status of DNGR-1. To make a clear distinction between the two conformational states, we henceforth refer to the reduction-sensitive form as "type-1 dimer" and the reduction-insensitive form as "type-2 dimer" (Fig 5C).

## Mutations in the neck region affect type-2 dimer formation and dimerization of DNGR-1

We observed type-2 dimer formation in both mouse and human DNGR-1 (see above), suggesting that the part of the neck region

involved in the process is conserved between the two. In order to pinpoint its location, we genetically removed overlapping blocks of 10–11 amino acids from the conserved part of the neck [K57–L66 (Δ1), L64–I73 (Δ2), L72–L82 (Δ3), N81–T90 (Δ4), R87–A96 (Δ5), and Q95–S104 (Δ6) (Fig 6A)], expressed the resulting constructs as soluble ECD proteins, and tested their ability to form type-2 dimers. The mutants devoid of the first two blocks (Δ1 and Δ2) were expressed comparably to the wild-type (WT) ECD and showed no phenotype with regard to type-2 dimer formation. The Δ3 mutant also displayed no obvious phenotype but expressed very poorly, while the Δ6 mutant failed to express at all (Fig 6B). Interestingly, the Δ4 mutant expressed as efficiently as the WT, but showed enhanced type-2 dimer formation even under neutral conditions (Fig 6B). The Δ5 mutant lacks the dimerization cysteine (C94) and consequently expressed as a monomer (Fig 6B). Unlike the C94S mutant, however, the Δ5 mutant expressed at levels comparable to the WT (Fig 6C).

To assess whether the effects observed with the deletion mutants were a result of the loss of specific amino acids or a non-specific consequence of segments of the neck region getting "out of sync" due to part of the helices being removed, we substituted all the residues within blocks 3, 4, and 6 with strings of alanines. Replacing all the residues in block 4 with alanine (Δ4A) resulted in an ECD protein that behaved like the Δ4 mutant in that it formed type-2 dimers in neutral conditions (Fig 6C). Interestingly, and unlike the Δ3 and Δ6 deletion mutants, the mutants with alanines in blocks 3 and 6 (Δ3A and Δ6A) were able to be expressed, although, in the case of Δ6A, to a lower extent than the WT ECD. Notably, while the Δ3A mutant showed no phenotype with respect to type-2 dimer formation, the Δ6A mutant failed to form type-2 dimers under conditions which were effective in inducing their formation in the WT protein (Fig 6C). Under non-reducing conditions, all mutants except the Δ5 ran as dimers (Fig 6C), confirming that the inability of the Δ6A ECD to form type-2 dimers is not because it lost the ability to make the disulfide bond.

We noticed a putative N-glycosylation site (NxT sequence) (Gavel & von Heijne, 1990) at the boundary of blocks 3 and 4, which is disrupted in both Δ3 and Δ4 mutants (Fig 6A). Consistent with this site being glycosylated in mouse DNGR-1, both Δ3A and Δ4A mutants showed slightly increased electrophoretic mobility compared to the WT ECD or the Δ6A mutant (Fig 6C). As Δ3A and Δ4A mutants do not exhibit concordant phenotypes, however, the glycosylation appears not to be involved in type-2 dimer formation.

Finally, to confirm that the phenotypes we observed for the ECD mutant proteins were applicable to the full-length transmembrane receptor, we expressed full-length WT, Δ4, and Δ6A DNGR-1 in cells. We lysed the cells in either 10 mM MES pH 6.1 or PBS-based buffers and analyzed the samples by reducing SDS–PAGE and Western blot with anti-DNGR-1 mAb. The WT receptor appeared exclusively as a type-1 dimer in the PBS sample, while lysis in the MES buffer induced type-2 dimer formation (Fig 6D). On the other hand, the Δ4 mutant appeared in the form of type-2 dimers in both buffers, while the Δ6A mutant exhibited no type-2 dimer formation in either buffer (Fig 6D). Thus, the data with full-length receptor recapitulate those obtained with ECD proteins. Interestingly, in the case of the type-2 dimer containing samples (WT in MES buffer and Δ4 mutant in both conditions), we reproducibly observed

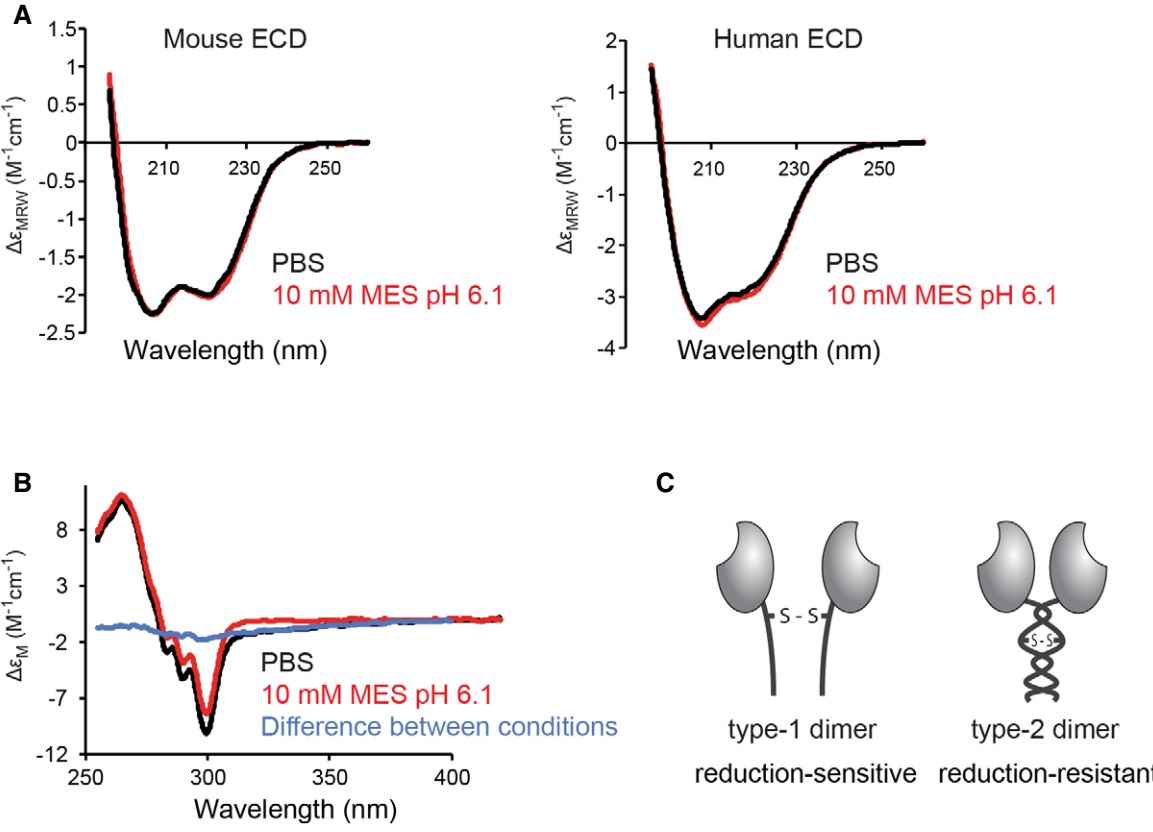

**Figure 5. A change in the tertiary structure is responsible for the formation of reduction-resistant dimers.**

A  *Long* mouse and human DNGR-1 ECD proteins were diluted in PBS or 10 mM MES pH 6.1 buffer and 20 independent far-UV spectra were acquired for each condition (red and black lines depict the composite curve for each condition).

B  *Long* mouse DNGR-1 ECD was diluted in PBS or 10 mM MES pH 6.1 buffer and 20 independent near-UV spectra were acquired for each condition (red and black lines depict the composite curve for each condition and the blue line shows the difference between the two).

C  Schematic representation of the suggested conformational states of DNGR-1.

**Table 2. Content of the elements of secondary structure in *long* mouse and human DNGR-1 ECD.**

|  | α-helix | β-sheet | β-turn | Random |
|---|---|---|---|---|
| *Long* mouse ECD |  |  |  |  |
| PBS | 16.1% | 22.5% | 17.1% | 44.3% |
| MES | 17.8% | 21.8% | 17.0% | 43.5% |
| Human ECD |  |  |  |  |
| PBS | 22.3% | 21.3% | 19.5% | 37.2% |
| MES | 24.2% | 20.7% | 20.8% | 34.2% |

multiple bands, seemingly corresponding to reduction-resistant oligomers (Fig 6D), indicating that, in the context of the full-length receptor, complexes of higher order than dimers are possible.

**Type-2 dimer formation does not affect the ability of DNGR-1 to bind F-actin, signal to NFAT, or undergo internalization**

A pH-induced conformational change in another C-type lectin receptor, DEC205, has recently been suggested to allow the protein to

recognize a ligand in apoptotic and necrotic cells (Cao *et al*, 2015). Consequently, we set out to determine whether the conformational change in DNGR-1 affects its ability to bind its ligand. To this end, we utilized a dot blot assay (Ahrens *et al*, 2012; Hanč *et al*, 2015) in which decreasing amounts of *in vitro* polymerized actin were spotted onto a membrane, which was then incubated with equal amounts of DNGR-1 WT, Δ4, or Δ6A mutant ECDs in PBS or MES pH 6.1 buffer. Binding was revealed with anti-FLAG antibody and strength of signal quantified by densitometry. Using this setup, we could not observe any differences in the binding ability of WT, Δ4, or Δ6A DNGR-1 ECDs in either buffer (Fig 7A), indicating that type-2 dimer switching is not involved in the regulation of DNGR-1 ligand binding. A slight trend toward lower binding in the MES buffer could be seen in some experiments, likely attributable to a small pH-induced conformational change in the actin filaments themselves (Oda *et al*, 2001).

Internalization of dectin-1 was previously shown to terminate signaling (Rosas *et al*, 2008; Hernanz-Falcón *et al*, 2009). The switch to the type-2 dimer conformation could conceivably allow similar regulation in the case of DNGR-1: repositioning of the neck regions could lead to repositioning of the intracellular signaling motifs, affecting interaction of DNGR-1 with Syk downstream. To

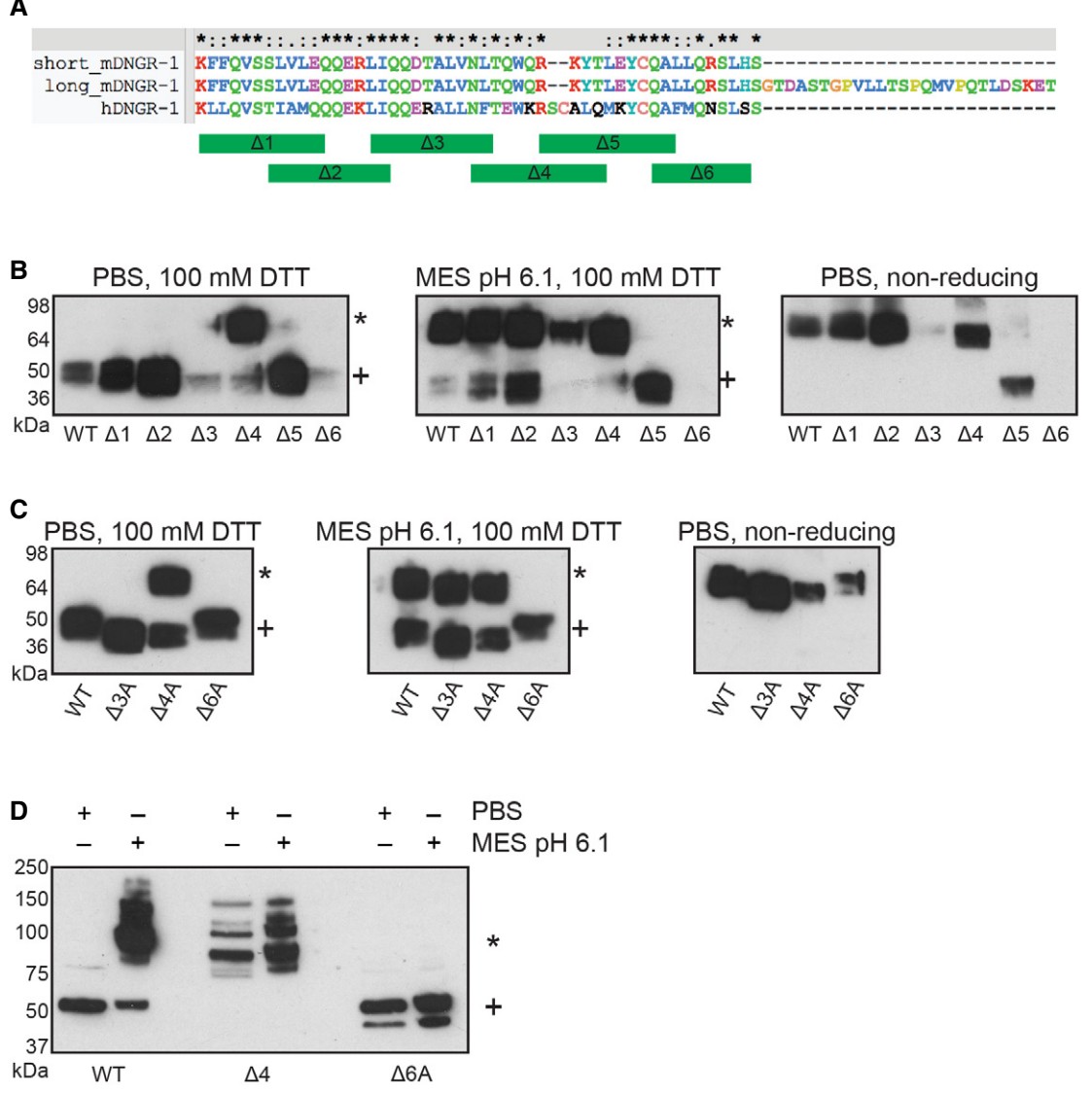

**Figure 6. Distinct parts of the neck region contribute to regulation of type-2 dimer formation.**

A    Sequences of the neck regions of human and *long* and *short* isoforms of mouse DNGR-1 aligned in Clustal X software. Parts of the neck corresponding to the block deletions are depicted as green bars.

B, C    Supernatants containing the indicated ECD proteins (mouse, *long* isoform) were diluted 1:2 into the indicated buffers and analyzed by SDS–PAGE and Western blot under reducing or non-reducing conditions. HRP-conjugated anti-FLAG antibody was used for detection of all proteins.

D    Phoenix cells expressing the indicated proteins were lysed in 1% SDS in PBS or 10 mM MES pH 6.1 buffers and the lysates were analyzed by SDS–PAGE and Western blot under reducing conditions. Anti-DNGR-1 antibody (clone 397) followed by HRP-conjugated secondary anti-rat antibody was used for detection.

Data information: Numbers on the side of blots indicate positions of molecular weight markers. Reduction-resistant dimers are indicated with an * and reduction-sensitive protein with a +.
Source data are available online for this figure.

test this hypothesis, we expressed the full-length WT, Δ4, and Δ6A DNGR-1 proteins in B3Z-Syk reporter cells, in which Syk activation is coupled to an NFAT reporter (Sancho *et al*, 2009). To assess whether the neck region as a whole is involved in DNGR-1 signaling, we also separately expressed chimeric proteins in which the entire neck region is replaced by that from two unrelated C-type lectins, CD69 and Ly49, which are also disulfide-bonded dimers (Sancho *et al*, 2000; Back *et al*, 2009). When treated with dead cells

as a source of F-actin, or when DNGR-1 was cross-linked with an antibody to induce signaling, we observed equivalent NFAT reporter activity in all the reporter cells expressing DNGR-1 mutant or WT proteins (Fig 7B). The cells expressing the CD69 chimera showed a decrease in NFAT activation when treated with dead cells but not with anti-DNGR-1 antibody (Fig 7C), suggesting that the chimera can still signal through Syk and the observed phenotype is due to low expression levels (data not shown). The Ly49 chimera induced

signaling comparable to that elicited by WT DNGR-1 (Fig 7C). In sum, these data indicate that type-2 dimer formation, and the exact structure of the neck region as a whole does not affect the ability of DNGR-1 to signal through Syk for NFAT activation.

Finally, after cross-linking by F-actin or anti-DNGR-1 antibody, WT DNGR-1 is rapidly internalized, which can be read out as a loss of surface staining by flow cytometry (Hanč *et al*, 2015). However, when cells expressing the full-length WT, Δ4, or Δ6A DNGR-1 mutants, or the chimeric proteins were tested for receptor internalization differences, none were observed (Fig 7D). Thus, type-2

dimer formation does not appear to be involved in regulation of endocytosis of surface DNGR-1.

### Type-2 dimer formation regulates DNGR-1 dependent cross-presentation in dendritic cells

The primary function of DNGR-1 in DCs is to facilitate cross-presentation of dead cell-associated antigens (Sancho *et al*, 2009; Iborra *et al*, 2012; Zelenay *et al*, 2012). To test whether the neck region and type-2 dimer formation are involved in this function, we resorted to

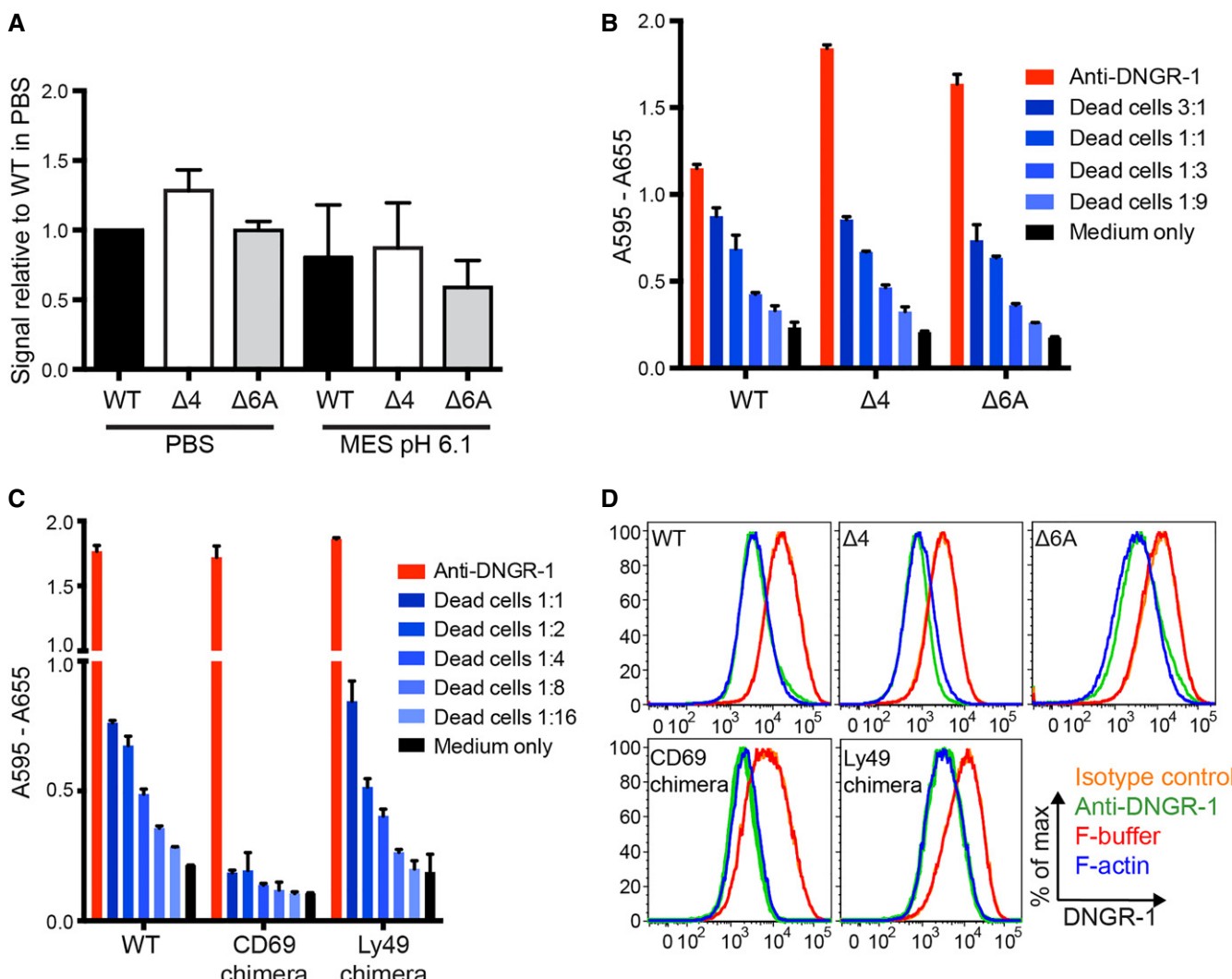

**Figure 7. Type-2 dimer formation does not affect DNGR-1 ligand binding, coupling to Syk or receptor internalization.**

A    Decreasing amounts of F-actin were spotted onto a nitrocellulose membrane and binding of indicated DNGR-1 ECD (mouse, *long* isoform) proteins was tested in PBS or MES pH 6.1 buffers. HRP-conjugated anti-FLAG antibody was used for detection. The signal was quantified by densitometry and quantitation of four independent experiments is shown. The bars represent mean ± SD.

B, C    B3Z-Syk reporter cells expressing indicated mutant or WT DNGR-1 receptors (mouse, *long* isoform) were incubated with UV-irradiated 293T cells (ratio of dead: reporter cells is indicated), or with plate-bound anti-DNGR-1 antibody or medium alone at 37°C overnight. Activation of the NFAT reporter was measured at the end of the incubation period. Data are plotted as mean ± SD of experimental duplicates. One representative of three experiments is shown.

D    Phoenix cells expressing indicated mutant or WT DNGR-1 proteins (mouse, *long* isoform) were treated with F-actin, F-actin buffer, anti-DNGR-1 antibody, or isotype-matched antibody of irrelevant specificity for 45 min, fixed in PFA, surface-stained for DNGR-1, and analyzed by flow cytometry. One representative of three experiments is shown.

the DC line MuTuDC1940 (MuTu), which, unlike other commonly used sources of DCs (e.g. GM-CSF-derived DCs), is clonal, expresses DNGR-1 (Fuertes Marraco et al, 2012), and can be used to study DNGR-1-dependent cross-presentation of dead cell-associated antigens (data not shown; see below). We ablated DNGR-1 in MuTu by CRISPR-Cas9 technology, introducing into the cells a doxycycline-inducible Cas9 (Wang et al, 2014) together with an sgRNA corresponding to a part of the first exon of DNGR-1. After induction of Cas9 expression with doxycycline followed by FACS sorting for DNGR-1-negative cells, we obtained a sub-clone of MuTu cells that had lost DNGR-1 expression (Fig 8A). After doxycycline withdrawal, expression of the wild-type or mutant receptors could be reconstituted using retroviral transduction, followed by cell sorting to obtain uniform and broadly similar levels of receptor expression (Fig 8A).

To test the ability of the KO and complemented MuTu sublines to mediate cross-presentation of dead cell-associated antigens, we cocultured them with UV-irradiated ovalbumin-expressing mouse embryonic fibroblasts (bm1 T OVA MEFs; Sancho et al, 2009) and pre-activated OVA-specific CD8$^+$ T cells (OT-I) overnight and measured the extent of T-cell reactivation by assessing the amount of IFN-$\gamma$ that accumulated in the culture medium. Consistent with previous data (Sancho et al, 2009; Iborra et al, 2012; Zelenay et al, 2012; Hanč et al, 2015), DNGR-1-deficient MuTu DCs showed impaired ability to cross-present dead cell-associated OVA when compared with WT cells (Fig 8B). Importantly, the phenotype of DNGR-1 KO cells could be rescued by reintroduction of WT DNGR-1, confirming that the effect is DNGR-1-dependent and not due to Cas9 off-target effects (Fig 8B). Interestingly, introduction of the Δ4 mutant into DNGR-1 KO MuTus resulted in increased

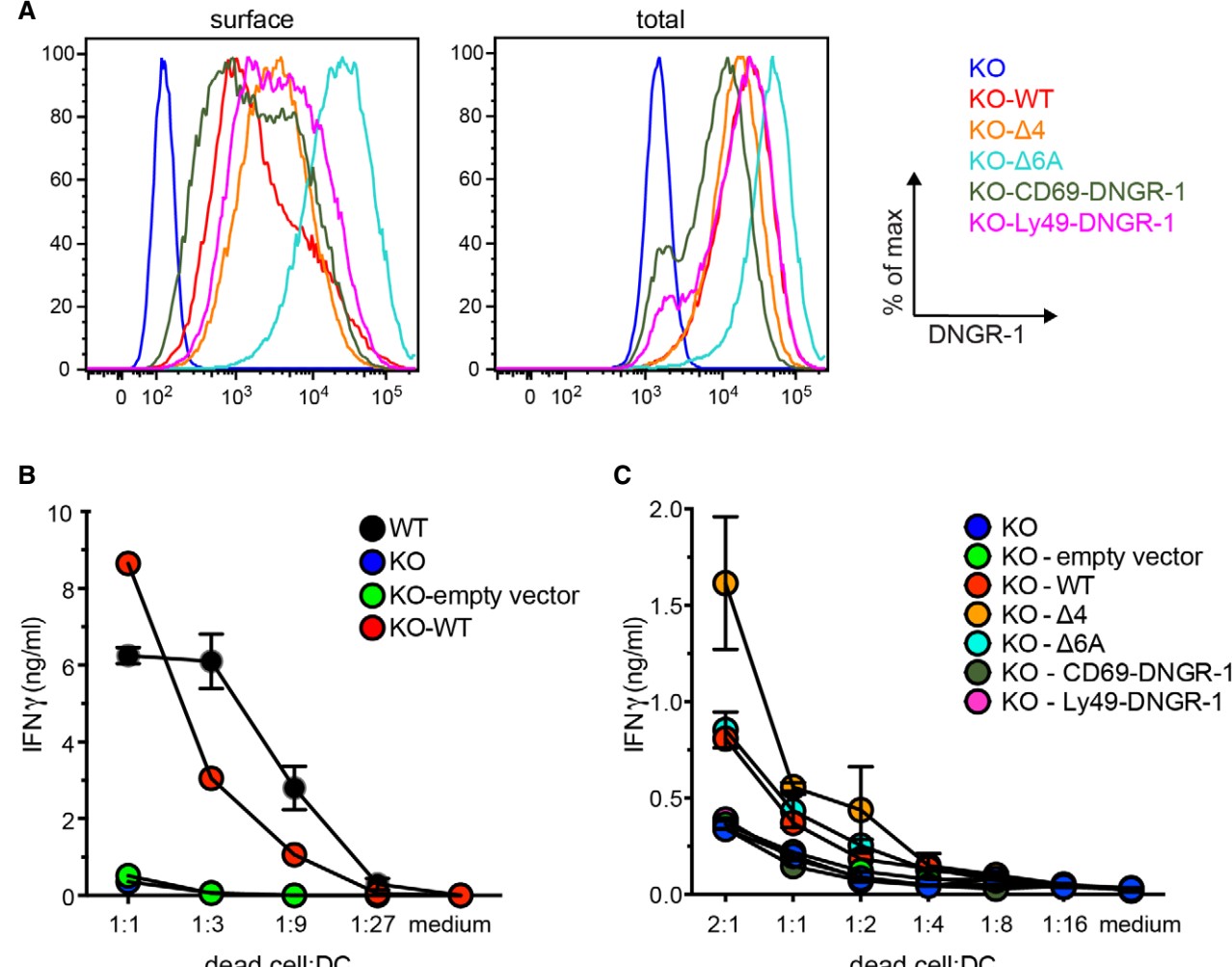

**Figure 8.   The neck region of DNGR-1 controls its ability to promote cross-presentation of dead cell-associated antigens.**

A    DNGR-1 KO MuTu DCs were transduced to express indicated DNGR-1 proteins, FACS-sorted, and expanded. Surface (left panel) and total (right panel) DNGR-1 expression was determined by flow cytometry.

B, C    Ovalbumin-expressing mouse embryonic fibroblasts (bm1 T OVA MEFs) were UV-irradiated, left to undergo secondary necrosis, and incubated overnight with MuTu cells transduced with indicated WT or mutant DNGR-1 proteins or an empty vector control (ratio of dead cells: MuTu cells is indicated) and OVA-specific pre-activated OT-I CD8$^+$ T lymphocytes. The amount of IFN-$\gamma$ accumulating in the culture medium was assessed using ELISA. One representative of two (B) or four (C) experiments is shown. Data are plotted as mean $\pm$ SD of an experimental duplicate.

amounts of IFN-γ accumulating in the medium after the incubation with dead cells (Fig 8C), despite, if anything, lower levels of expression than the WT receptor (Fig 8A). In contrast, chimeric DNGR-1 containing neck regions of CD69 and Ly49 failed to rescue the phenotype, despite comparable levels of expression to the WT receptor. Finally, the Δ6A mutant phenocopied the WT receptor (Fig 8C). Taken together, our data suggest that the integrity of the neck region of DNGR-1 is crucial for the function of the receptor in promoting cross-presentation of dead cell-associated antigens by DCs and that the switch to type-2 dimer conformation, which is enhanced in the Δ4 mutant, allows for higher efficiency of the process.

## Discussion

DNGR-1 is a sensor of extracellular F-actin that allows DCs to detect the presence of cell debris (Sancho *et al*, 2009). Engagement of DNGR-1 does not result in activation of DCs but modulates the fate of DC endosomes containing dead cell debris and improves cross-presentation of antigens associated with those debris (Sancho *et al*, 2009; Iborra *et al*, 2012; Zelenay *et al*, 2012) to CD8⁺ T cells. The molecular mechanism through which DNGR-1 favors cross-presentation of dead cell-associated antigens remains unknown but is likely to depend on the properties of the receptor, including the ligand-binding CTLD, the transmembrane portion and cytosolic signaling domain, and, critically, the neck region that couples the CTLD to the transmembrane domain. Here, we show that the neck region of DNGR-1 serves as a pH- and ionic strength-specific sensor that undergoes a conformational change in a physiologically relevant range of conditions. The conformational change happens predominantly at the level of tertiary structure, suggesting a rearrangement of the α-helices that likely form the majority of the neck region of the receptor based on molecular modeling (P. Hanč & C. Reis e Sousa, unpublished observations). pH-induced conformational changes have been reported for other proteins and are often mediated by protonation of one or more histidines (Krukenberg *et al*, 2009; Harrison *et al*, 2013; Kalani *et al*, 2013; Dai *et al*, 2014; Cao *et al*, 2015) or, under certain circumstances, of negatively charged residues (Yeo *et al*, 2014), leading to the creation of local electrostatic alterations. However, the DNGR-1 neck region does not bear conserved histidines or patches of negatively charged residues. On the other hand, it contains multiple hydrophobic residues and residues potentially capable of forming hydrogen bonds and salt bridges. While neither of these interactions *per se* can provide (de) stabilizing forces in the same range as electrostatic repulsion, they can contribute to, and perhaps even be sufficient for, pH-sensitive conformational rearrangements (Harrison *et al*, 2013). Through mutagenesis of the conserved part of the neck, we identified proteins that are either locked in (Δ4) or unable to switch to (Δ6A) the reduction-insensitive state. Importantly, the inability of the Δ6A mutant to form reduction-insensitive dimers could either be due to a genuine inability to undergo the conformational change or due to removal of a part of the protein that mediates protection of the disulfide bond. These are two scenarios that cannot be differentiated by our biochemical assay although the observed lack of phenotype of the Δ6A mutant in all functional assays may suggest that block 6 primarily serves to protect the disulfide bond and its removal results in a protein that is still able to undergo the conformational change

but is now reduction sensitive. In contrast, the data obtained with the Δ4 mutant are most consistent with a model in which residues in block 4 serve as a spring, "pushing" the neck regions apart and preventing the switch to the type-2 dimer conformation under neutral pH conditions.

Interestingly, while DNGR-1 soluble ECDs predominantly form type-2 dimers at low pH/ionic strength, the full-length receptor expressed in cells additionally forms higher order reduction-resistant oligomers under the same conditions. Reduction-sensitive oligomers can also be detected as a minor component in ECD preparations but these likely correspond to a minor fraction of protein that failed to fold properly during biosynthesis as they do not respond to changes in the pH and ionic strength and can be removed by size exclusion chromatography. Overall, these observations suggest that regions of the receptor not present in the ECD—in particular the transmembrane and intracellular portions—could contribute to the formation of reduction-resistant complexes. Albeit uncommon, disulfide bond formation has previously been reported for intracellular proteins, especially in the context of oxidative stress (Cumming *et al*, 2004). Similarly, formation of disulfide bonds in transmembrane regions of integral membrane proteins can also be induced, at least *in vitro* (Lynch & Koshland, 1991). Consequently, the reduction-resistant higher oligomers could represent physiologically relevant forms of DNGR-1 whose role remains to be established.

Binding to F-actin, internalization from the cell surface, and interaction with Syk kinase are all integral to DNGR-1 function (Sancho *et al*, 2009; Iborra *et al*, 2012; Zelenay *et al*, 2012; Hanč *et al*, 2015). The conformational change does not appear to impact receptor endocytosis from the cell surface, affect Syk activation in a reporter cell line, or modify receptor binding to F-actin. These data would suggest that the conformational change could be important for other aspects of DNGR-1 biology that ultimately impact its natural function in promotion of cross-presentation. Our data with mutant proteins in a reconstituted overexpression system are consistent with that notion but need to be confirmed by expression from the endogenous DNGR-1 locus. Similarly, the exact role of the conformational change is exceedingly difficult to assess at present as the molecular mechanism underlying cross-presentation remains unknown 40 years after discovery of the phenomenon (Bevan, 1976). Nevertheless, it is interesting to note that the conditions in which DNGR-1 is found in the type-1 dimer conformation correspond to the conditions in the extracellular milieu. On the other hand, conditions present in endosomes would be predicted to induce a switch to the type-2 dimer conformation (Scott & Gruenberg, 2011). We suppose that the pool of DNGR-1 reaching late endocytic compartments, with lowest pH, may be most prone to undergo type-2 dimer switching. However, the switch could start even in early endosomes where the concentration of salt ions does not exceed 150 mM and the pH already drops to 6.2 (Scott & Gruenberg, 2011), conditions that can efficiently induce the reduction-insensitive conformation *in vitro* (Fig 2B). It is, however, important to note that the exact compartment in which DNGR-1 undergoes the conformational change cannot be identified by biochemical analysis because the conformational state of DNGR-1 is reversible and dictated by the buffer conditions of extraction rather than those of the endosomes from which it is extracted (see Fig 6D). As such, analysis of DNGR-1 conformers within the endocytic pathway will

likely require development of imaging probes compatible with live-cell microscopy.

In sum, in this report we show that the neck region of DNGR-1 can serve as a sensor of local pH and ionic strength and can cause the receptor to undergo a reversible conformational switch, likely mediated by mutual repositioning of the two neck regions within the DNGR-1 dimer. Notably, the conformation state appears to influence receptor function independently of internalization, ligand binding, or Syk signaling. The mechanistic explanation of these observations awaits understanding of the mechanism by which DNGR-1 mediates extraction of antigenic material from dead cell debris for cross-presentation. Importantly, while we demonstrate that the integrity of the neck region is necessary for the function of DNGR-1, the neck region alone is not sufficient. Indeed, targeting antigens to a chimeric protein consisting of the neck region of DNGR-1 and the intracellular, transmembrane, and ligand-binding domains of dectin-1 did not result in an increase in cross-presentation (data not shown). Thus, other parts of DNGR-1 are also involved in determining the ability of the receptor to promote cross-presentation of dead cell-associated antigens.

# Materials and Methods

### ECD proteins

DNA coding for WT or mutant proteins corresponding to the extracellular domain of DNGR-1 (K57–I264 for *long* mouse, K57–I238 for *short* mouse, and K57–V241 for the human isoform) or mouse dectin-1 (R71–L244) was inserted into the p3xFLAG-CMV-9 expression vector (Sigma-Aldrich) and all the constructs were verified by sequencing. The proteins were expressed by transient transfection in 239F cells, as described previously (Ahrens *et al*, 2012; Hanč *et al*, 2015) and, where indicated, purified by affinity chromatography on M2 anti-FLAG matrix (Sigma-Aldrich) followed by size exclusion chromatography on Superdex S200 column (GE Healthcare).

### DNGR-1 mutagenesis

All mutants were prepared using QuikChange Lightning kit (Agilent Technologies) and following primers: C94S: 5′-CCATGCATGATCCA ATTAGGAAGGCATGTTTCCTTGCTGTCCAGGGTCTG-3′, Δ1: 5′-CA AGGATGACGATGACAAGCTTGCGGCCGCGGAGCAGCAGGAAAGAC TCATCCAACAGGAC-3′, Δ2: 5′-CGGCCGCGAAGTTCTTCCAGGTAT CCTCTCAACAGGACACAGCATTGGTGAACCTTACAC-3′, Δ3: 5′-CC TCTCTTGTCTTGGAGCAGCAGGAAAGAACACAGTGGCAGAGGAAA TACACACTGGAATACTGCC-3′, Δ3A: 5′-GGTATCCTCTCTTGTCTTG GAGCAGCAGGAAAGAGCCGCCGCCGCCGCCGCCGCCGCCGCCGCC GCCACACAGTGGCAGAGGAAATACACACTGGAATACTGCCAAGCC-3′, Δ4: 5′-GGAAAGACTCATCCAACAGGACACAGCATTGGTGCTGGAA TACTGCCAAGCCTTACTGCAGAGATCTCTCC-3′, Δ4A: 5′-GCAGGA AAGACTCATCCAACAGGACACAGCATTGGTGGCCGCCGCCGCCGCC GCCGCCGCCGCCCTGGAATACTGCCAAGCCTTACTGCAGAG-3′, Δ5: 5′-GACACAGCATTGGTGAACCTTACACAGTGGCAGTTACTGCA GAGATCTCTCCATTCAGGCAC-3′, Δ6: 5′-CAGTGGCAGAGGAAATA CACACTGGAATACTGCGGCACAGATGCTTCTACTGGACCAG-3′, Δ6A: 5′-GTGAACCTTACACAGTGGCAGAGGAAATACACACTGGAATACT

GCGCCGCCGCCGCCGCCGCCGCCGCCGCCGGCACAGATGCTTC TACTGGACCAGTTCTTCTGAC-3′.

For the chimeric DNGR-1:dectin-1 protein, a "megaprimer" covering the whole neck region of *long* mouse DNGR-1 (K57–T130) with 5′ and 3′ sequences complementary to regions of the p3xFLAG-CMV-9 plasmid and dectin-1, respectively, was amplified in a PCR with Phusion Hot Start II polymerase (Thermo Scientific), using the following primers: 5′-GCTGGGTGCCCTAGCATTTTGGAAGTTCTTC CAGGTATCCTCTCTTG-3′ and 5′-CCATGCATGATCCAATTAGGAA GGCATGTTTCCTTGCTGTCCAGGGTCTG-3′ and a DNGR-1-coding plasmid as a template, and then used in a modified reaction with QuikChange Lightning kit (Agilent Technologies) as per manufacturer's instructions with p3xFLAG-CMV-9 plasmid containing dectin-1 ECD sequence as a template. For generation of chimeric DNGR-1 proteins with the neck region of CD69 or Ly49, "megaprimers" covering the entire neck region of appropriate proteins were generated using the following primers: 5′-GTTAGCAACGTCCAT TTTCTTGGGCATCGGCAAGTACAATTGCCCAGGCTTG-3′ and 5′-G TGGACAAGGGCTGCAGTCACTACCAGCAACATGGTGGTCAGATG-3′ for CD69 and 5′-GTTAGCAACGTCCATTTTCTTGGGCATCAACA TTTTTCAGAATAGTCAACAAAATCATGAACTGCAGG-3′ and 5′-GT GGACAAGGGCTGCAGTCACTACCTTCAAAACCTCTGCCTGTGTGCT GTGAGG-3′ for Ly49. cDNA generated from total splenic mRNA of a C57BL/6 and a CBA/J mouse, respectively, was used as templates. "Megaprimers" were then used in a modified reaction with Quik-Change Lightning kit (Agilent Technologies) as per manufacturer's instructions with pFB-IRES GFP plasmid containing the entire sequence of DNGR-1 as a template.

### SDS–PAGE and Western blot analysis

Samples for SDS–PAGE were prepared in 6× Laemmli buffer (60% (v/v) glycerol, 150 mg/ml SDS, 0.75 mg/ml bromophenol blue in 75 mM Tris–HCl pH 6.8) under reducing (+100 mM DTT) or non-reducing conditions unless otherwise stated. Separation was carried out using 4–20% Mini-PROTEAN TGX precast gels (Bio-Rad). Proteins were transferred onto Immobilon P membrane (Merck Millipore) using wet transfer, the membrane was blocked in 5% milk in PBS + 0.05% Tween-20, and ECD proteins were detected using HRP-conjugated M2 anti-FLAG antibody (Sigma-Aldrich). Full-length DNGR-1 was detected using anti-DNGR-1 antibody (clone 397) followed by HRP-conjugated polyclonal anti-rat antibody (IgG (H+L); Stratatech).

### Circular dichroism measurements

For far-UV CD, mouse *long* DNGR-1 ECD was concentrated to 7 mg/ml and, immediately before analysis, diluted into PBS or 10 mM MES pH 6.1 to final concentration 170 μg/ml. Human DNGR-1 ECD was concentrated to 1.4 mg/ml and immediately before analysis diluted into PBS or 10 mM MES pH 6.1 to final concentration 140 μg/ml. Far-UV CD spectra (260–195 nm) were recorded at 20°C in 1 mm fused silica cuvettes using a Jasco J-815 spectropolarimeter fitted with a cell holder temperature controlled by a CDF-426S Peltier unit. The spectra were typically recorded with 0.1-nm resolution and baseline corrected by subtraction of the appropriate buffer spectrum. CD intensities are presented as the CD absorption coefficient calculated on a mean residue weight basis ($\Delta\varepsilon_{MRW}$). Secondary

structure content was estimated using methods originally described by Sreerama and Woody (Sreerama & Woody, 2000). For near-UV CD, mouse *long* DNGR-1 ECD was concentrated to 5 mg/ml and, immediately before analysis, diluted tenfold into PBS or 10 mM MES pH 6.1. Near-UV CD spectra (450–255 nm) were recorded at 20°C in 10 mm fused silica cuvettes. Near-UV CD intensities are presented on a molecular weight basis ($\Delta\varepsilon_M$).

**Dot blot assay**

Dot blot assay was performed as described previously (Ahrens *et al*, 2012; Hanč *et al*, 2015). Briefly, lyophilized human platelet actin (Cytoskeleton) was reconstituted in G-buffer (5 mM Tris–HCl, pH 8.0 + 0.2 mM $CaCl_2$) and polymerized in F-buffer (10 mM Tris–HCl, pH 7.5 + 50 mM KCl + 2 mM $MgCl_2$ + 1 mM ATP). Polymerized actin was stabilized with 5 μM phalloidin (Life Technologies) and spotted onto nitrocellulose membrane (Whatman) in twofold dilution series. The membrane was blocked in 5% milk in PBS + 0.05% Tween-20 overnight and incubated with protein supernatants diluted into appropriate buffers at equal concentrations. Binding of all proteins was detected using HRP-conjugated M2 anti-FLAG antibody (Sigma-Aldrich) and revealed using SuperSignal West Pico Chemiluminescent Substrate (Thermo Scientific) or Luminata Forte Western HRP Substrate (Merck Millipore).

**Cells**

Phoenix, B3Z-Syk (Sancho *et al*, 2009), and GP2-293 cells were grown in RPMI1640 medium (Gibco) supplemented with glutamine, penicillin, streptomycin, β-ME, and 10% heat-inactivated fetal calf serum (FCS) at 37°C and 5% $CO_2$. MuTu DC1940 cells were grown in IMDM medium (Gibco) supplemented with β-ME and 10% heat-inactivated fetal calf serum (FCS) at 37°C and 5% $CO_2$. 293F cells were grown in FreeStyle 293 Expression Medium (Gibco) at 37°C, 8% $CO_2$, and with constant shaking on an orbital shaker at 120 rpm. 293FT cells were grown in DMEM medium (Gibco) supplemented with 10% heat-inactivated fetal calf serum (FCS) at 37°C and 10% $CO_2$.

**Retroviral transduction**

GP2-293 packaging cells were transfected with a mixture of Gene-Juice (Novagen), VSV-G envelope protein-coding plasmid, and a pFB plasmid coding for the desired protein. On days 1, 2, and 3 post-transfection, the pseudotyped virus-containing culture medium was recovered, filtered, supplemented with 8 μg/ml polybrene (Sigma-Aldrich), and immediately applied to target cells. The plate was centrifuged for 90 min at 2,500 × *g* at room temperature and left in the incubator for a further 90 min. After the incubation, the medium was exchanged for fresh complete RPMI1640 medium. For transduction of B3Z-Syk cells and MuTu DC1940 cells, ecotropic virus-containing supernatant from Phoenix cells was used.

**DNGR-1 KO MuTu line generation**

293FT cells were transfected by a mixture of Lipofectamine 2000 (Thermo Fisher Scientific), psPax2, pMD2.G, and pCW-Cas9 plasmids. Lentivirus-containing culture medium was recovered on days

1, 2, and 3 post-transfection, filtered, diluted 1:9, supplemented with 8 μg/ml polybrene (Sigma-Aldrich), and immediately applied to MuTu cells. The plate was centrifuged for 90 min at 2,500 × *g* at room temperature and left in the incubator for a further 90 min. After the incubation, the medium was exchanged for fresh complete IMDM medium. On day 5, selection medium containing 0.5 μg/ml puromycin was applied. Surviving cells after 1 week of selection were expanded and transduced in an identical manner as described above with lentivirus generated in 293FT cells transfected with a mixture of Lipofectamine 2000, psPax2, pMD2.G, and pLX-sgRNA plasmids. Successfully transduced cells were selected in medium containing 5 μg/ml blasticidin. Surviving cells after 1 week of selection were expanded and Cas-9 expression was induced by addition of 1 μg/ml doxycycline to the culture medium for 3 weeks. Cells that lost DNGR-1 expression were then purified by FACS sorting. The sequence of sgRNA used was as follows: CTGAACATTTGCTAGGGGAT. pCW-Cas9 and pLX-sgRNA plasmids were a gift from Eric Lander & David Sabatini (Addgene plasmids #50661 and #50662), pMD2.G and psPAX2 plasmids were a gift from Didier Trono (Addgene plasmids #12259 and #12260).

**Internalization assay**

Internalization assay was described previously (Hanč *et al*, 2015). Briefly, cells expressing the indicated full-length DNGR-1 proteins were treated with either 1 μM F-actin, F-buffer, 10 μg/ml anti-DNGR-1 (clone 7H11), or 10 μg/ml isotype control antibody and incubated at 37°C and 5% $CO_2$ for 45 min. After the incubation, the cells were harvested, washed in ice-cold PBS + 5 mM EDTA, and fixed in 4% PFA. After fixation, the cells were washed in FACS buffer (5 mM EDTA, 1% FCS, 0.0125% $NaN_3$) and surface-stained with PE-conjugated anti-DNGR-1 antibody (clone 1F6). Staining was analyzed using LSR Fortessa Flow cytometer (BD Biosciences). Data analysis was performed in FlowJo 9.8.5 software (TreeStar).

**B3Z-Syk reporter assay**

B3Z-Syk reporter assay was described previously (Sancho *et al*, 2009). Briefly, $1 \times 10^5$ B3Z-Syk reporter cells expressing indicated DNGR-1 WT or mutant proteins were plated with UV-irradiated (240 $mJ/cm^2$) 293T cells at the indicated ratios, or in wells coated with anti-DNGR-1 antibody (clone 7H11) or in medium alone in a 96-well plate. LacZ activity after incubation overnight at 37°C and 5% $CO_2$ was determined by lysing the cells in CPRG (Roche) containing buffer and measuring $OD_{595}$ with $OD_{655}$ as a reference at multiple time points.

**Cross-presentation assay**

Ovalbumin-expressing mouse embryonic fibroblasts (bm1 T OVA MEFs; Sancho *et al*, 2009) were UV-irradiated (UVC 240 $mJ/cm^2$) and left overnight to undergo secondary necrosis. Cell corpses were harvested, washed once in full RPMI1640 medium, and plated at indicated ratios with $6 \times 10^4$ MuTu cells per well in a U-bottom 96-well plate. After a 4-h incubation at 37°C and 5% $CO_2$, $3 \times 10^4$ pre-activated OT-I T cells (see below) were added to each well and incubated at 37°C and 5% $CO_2$ overnight. Next morning, the plate

was freeze–thawed once, and total amount of IFN-γ in the supernatants was determined by ELISA.

### OT-I cells

OT-I Rag1$^{-/-}$ mice were bred at The Francis Crick Institute under specific pathogen-free conditions in accordance with the national and institutional guidelines for animal care and approval by The Francis Crick Institute Animal Ethics Committee and by the Home Office, UK. Spleen and lymph nodes from one OT-I Rag1$^{-/-}$ mouse were homogenized into a single cell suspension. Cells were plated at $1 \times 10^5$ per ml in complete RPMI1640 medium supplemented with 0.1 nM SIINFEKL peptide. On day 3 of culture, IL-2 was added to a final concentration of 25 U/ml. CD8$^+$ T cells were MACS-enriched on day 5 and used immediately.

**Expanded View** for this article is available online.

### Acknowledgements

We thank all members of the Immunobiology Laboratory for helpful discussions and suggestions. We thank Hans Acha-Orbea for the kind gift of MuTu cells and The Francis Crick Institute Flow Cytometry, Equipment Park, and Structural Biology Science Technology Platforms for assistance. This work was supported by The Francis Crick Institute, which receives its core funding from Cancer Research UK (FC001136), the UK Medical Research Council (FC001136), and the Wellcome Trust (FC001136), as well as by a prize from Fondation Bettencourt-Schueller, and an Advanced Researcher grant (AdG 268670) from the European Research Council to CRS.

### Author contributions

PH designed the study, performed and analyzed experiments, and wrote the manuscript. OS provided technical assistance, contributed to the design of experiments, and performed some of the experiments shown in Fig 8. HF provided assistance in generating reagents. SRM performed and analyzed the experiments shown in Fig 5. SK provided technical assistance and contributed to the design of experiments. CRS designed and coordinated the study and wrote the manuscript.

### Conflict of interest

The authors declare that they have no conflict of interest.

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
