## [Review Process File · The EMBO Journal]

Manuscript EMBO-2016-94695

A pH- and ionic strength-dependent conformational change in the neck region regulates DNGR-1 function in dendritic cells

Pavel Hanc, Oliver Schulz, Hanna Fischbach, Stephen R Martin, Svend Kjaer, Caetano Reis e Sousa

Corresponding author: Caetano Reis e Sousa, The Francis Crick Institute

Review timeline:

Submission date:	03 May 2016
Editorial Decision:	10 June 2016
Revision received:	18 August 2016
Accepted:	15 September 2016
Accepted:	15 September 2016

Editor: Karin Dumstrei

Transaction Report:

1st Editorial Decision

10 June 2016

Thank you for submitting your manuscript to The EMBO Journal. Your study has now been seen by two referees and their comments are provided below.

As you can see there is an interest in the analysis. However, both referees also find that the findings need to be significantly extended to consider publication here. Both referees find that you need some more mechanistic insight into how the dimeric state of DNGR-1 affects cross presentation. Referee #2 also suggests an interesting experiment to further support the importance of the neck domain in facilitating cross-presentation. Should you be able to extend the findings and add more mechanistic insight then I am open to consider a revised version. I am aware that this might not be a straightforward task and if you are not able to do so then it is in your best interest to seek publication elsewhere at this stage.

When preparing your letter of response to the referees' comments, please bear in mind that this will form part of the Review Process File, and will therefore be available online to the community. For more details on our Transparent Editorial Process, please visit our website:
http://emboj.embopress.org/about#Transparent_Process

Let me know if we need to discuss things further.

REFeree REPORTS

Referee #1:

In the manuscript, Hanc et al. found that DNGR-1 can dimerize through the neck region of DNGR-1 monomer. Interestingly, they found that the DNGR-1 dimer had two forms. The "type-1 dimer" exists under regular PBS conditions and is reduction-sensitive, while the "type-2 dimer" forms at low pH or ionic strength conditions and becomes reduction-insensitive, and the transition between the two dimeric forms is reversible. They also found that the type-2 dimer had similar activities in F-actin binding and internalization, but could affect the cross-presentation in dendritic cells. Overall the findings are interesting, but the mechanisms for dimerization/oligomerization and cross-presentation are not quite clear.

Major:

1) The type-2 dimer can be formed under low pH and low ionic strength conditions, are the low ionic strength conditions physiologically relevant in the DNGR-1 pathway? Authors suggest that the reduction-insensitive oligomers might be the physiologically relevant forms of DNGR-1, would it be possible to test this in vitro using the full-length proteins? In Fig. 6D, $\Delta 6A$ only shows type-1 interactions (reduction sensitive), does this mean that the transmembrane and cytoplasm domain of DNGR-1 may not be involved in the reduction-insensitive dimerization or oligomerization? If this is the case, then why the ECD of DNGR-1 only shows the reduction-insensitive dimer instead of oligomer in vitro? The neck region mutants, such as $\Delta 4$ and $\Delta 6A$, do provide some information for DNGR-1 dimerization/oligomerization, but the mechanism of the neck interactions seems still confusing and probably needs to be explored by more experiments.

2) Is the regulation of cross-presentation by DNGR-1 type-2 dimer actin dependent? Have authors tested the role of actin in the regulation? According to the biochemical data in the manuscript, $\Delta 6A$ only forms type 1 dimer, WT can form both type 1 and type 2 dimer. But in Fig. 8C, KO- $\Delta 6A$ and KO-WT show similar effects in DC cross-presentation, so how to explain this result? It looks like the oligomeric states of DNGR-1 may affect the cross-presentation, but authors may need to provide more evidence on this, and also investigate the mechanism.

Referee #2:

The ectodomain (ECD + neck) of mouse and human DNGR-1 is made of glycosylated disulphide-bonded dimers. Interestingly, when DNGR-1 ectodomain were subjected to buffers of lower pH and ionic strength, they form reduction resistant dimers when reduction sensitivity was analysed by reducing SDS-PAGE and Western blot. 8 M urea treatment abolished the ability of DNGR-1 to resist reduction. This behaviour was found reversible and the human and mouse DNGR1 necks thought to be responsible for the reduction insensitivity of DNGR-1 dimers. In support of that view, grafting of the DNGR1 neck in place of that found in Dectin-1 induced a reduction-insensitive state in the Dectin-1 ECD. Further analysis of near- and far-UV CD spectra suggested that mutual repositioning of the two neck regions within the dimer was responsible for the observed reduction resistance and for the existence of reduction sensitive ("type-1 dimer") and reduction insensitive ("type-2 dimer") forms. Structure-function analysis of the neck showed that a mutant called N81 - T90 ($\Delta 4$) displays enhanced type-2 dimer formation even under neutral conditions, an observation that also stands when DNGR1 is expressed in its physiological - membrane bound - context. In view of the recent description of pH-induced conformational change in the C-type lectin receptor DEC205, the authors showed next that type-2 dimer formation does not affect the ability of DNGR-1 to bind F-actin, signal to NFAT and undergo internalization. Finally, the authors used the DC line MuTu to assess DNGR-1-dependent cross-presentation of dead cell-associated antigens to OT-I. Introduction of the $\Delta 4$ mutant into DNGR-1 KO MuTu resulted in a small (less than two-fold) increase in the amounts of IFN γ accumulating in the medium after the incubation with dead cells as compared to MuTu cells expressing WT DNGR-1. Therefore, the switch to type-2 dimer conformation, which is exacerbated in the $\Delta 4$ mutant, slightly enhanced the efficiency of the process. As discussed by the authors, conditions present in late endosomes and lysosomes would be predicted to induce a switch to the type-2 dimer conformation accounting the unique routing of DNGR-1 toward cross-presenting compartment. They fairly propose a few lines of experiments

aiming at explaining the mode of action of the DNGR-1 neck and relating it to cross-presentation To solidify their hypothesis the authors may have easily used the "fast track" MuTu expression system to express their Dectin-1 construct with a grafted DNGR1 neck. If their hypothesis is correct (that is the DNGR-1 neck is a evolutionary conserved and autonomous functional module), this will have allow them to readily test in a couple of months whether it confers the chimeric Dectin-1 molecules the de novo to capacity to cross-present native OVA targeted to Dectin-1 via an antibody conjugate.

Specific questions

Figure 1. Following reduction the long isoform 'band' appears composed of two bands. This needs to be discussed. Based on Figure 1C, they do not appear to correspond to distinct glycoforms. How are the few multimers assembled (see also Figure 6D)? Is there any additional Cys in the neck ?
Figure 7. Is it necessary to use pre-activated OVA-specific CD8+ T-cells (OT-I) overnight ?

1st Revision - authors' response

18 August 2016

Reviewer 1

The type-2 dimer can be formed under low pH and low ionic strength conditions, are the low ionic strength conditions physiologically relevant in the DNGR-1 pathway?

We thank the reviewer for pointing out that this has not been clearly stated in the manuscript. The exact ionic strength, especially in the later endocytic compartments is not well described, but for the early endosomes, the combined concentration of salt ions does not appear to exceed 150mM while the pH drops to around 6.2 (Scott et al. Bioessays, 2011). As can be seen in figure 2B, these conditions effectively induce the switch to the reduction-insensitive conformation. We have now added this clarification to the discussion (highlighted in yellow).

Authors suggest that the reduction-insensitive oligomers might be the physiologically relevant forms of DNGR-1, would it be possible to test this in vitro using the full-length proteins?

Unfortunately, full-length transmembrane proteins are not easily amenable to recombinant expression, precluding the experiment that the reviewer is suggesting.

In Fig.6D, $\Delta 6A$ only shows type-1 interactions (reduction sensitive), does this mean that the transmembrane and cytoplasm domain of DNGR-1 may not be involved in the reduction-insensitive dimerization or oligomerization? If this is the case, then why the ECD of DNGR-1 only shows the reduction-insensitive dimer instead of oligomer in vitro?

We think that the $\Delta 6A$ mutant is still forming the oligomers, but using our biochemical assay we do not see it because the dimer itself is no longer reduction-resistant, so the whole oligomeric complex falls apart in the presence of reducing agents.

The neck region mutants, such as $\Delta 4$ and $\Delta 6A$, do provide some information for DNGR-1 dimerization/oligomerization, but the mechanism of the neck interactions seems still confusing and probably needs to be explored by more experiments.

The mechanism of neck interactions is indeed unclear and a detailed description of the forces involved in the conformational change would first require a knowledge of the atomic structure of the neck region. We attempted to crystalize the whole ECD of DNGR-1 but, presumably due to the inherent flexibility of the neck region (see Hanc et al. Immunity, 2015) we never obtained diffracting crystals. There might be other ways to explore the different conformational states through NMR and/or protein structure modelling, but we believe this is beyond the scope of the current study.

Is the regulation of cross-presentation by DNGR-1 type-2 dimer actin dependent? Have authors tested the role of actin in the regulation?

Yes, DNGR-1-mediated cross-presentation, as well as the internalization of the receptor is F-actin dependent. See Hanc et al. Immunity, 2015.

According to the biochemical data in the manuscript, $\Delta 6A$ only forms type 1 dimer, WT can form both type 1 and type 2 dimer. But in Fig. 8C, KO- $\Delta 6A$ and KO-WT show similar effects in DC cross-presentation, so how to explain this result?

As we mentioned in the discussion, the biochemical assay that we use to observe the conformational change relies on the fact that part of the neck region protects the disulfide bond from the effects of reducing agents in one of the conformational states. Apparent loss of type 2 dimer formation in the biochemical assay can be caused by two factors – either the ability of the protein to undergo the conformational change has indeed been compromised, or, alternatively, the part responsible for the protection of the disulfide bond has been removed while the ability to undergo the conformational change itself remains unaffected. Based on the fact that the KO-WT and KO- Δ 6A cells show similar ability to cross-present, as the reviewer points out, we suggest that the latter is the case for Δ 6A. We have now attempted to clarify this further in the discussion (highlighted in yellow).

It looks like the oligomeric states of DNGR-1 may affect the cross-presentation, but authors may need to provide more evidence on this, and also investigate the mechanism.

The mechanism by which dendritic cells and other cell types cross-present exogenous antigens on MHC class I molecules remains poorly understood. Consequently, it is next to impossible to assess how the formation of Type-2 dimers and other DNGR-1 oligomers affects such mechanism. This is a focus of ongoing investigation, which we feel is beyond the scope of this study.

Reviewer 2

To solidify their hypothesis the authors may have easily used the "fast track" MuTu expression system to express their Dectin-1 construct with a grafted DNGR1 neck. If their hypothesis is correct (that is the DNGR-1 neck is a evolutionary conserved and autonomous functional module), this will have allow them to readily test in a couple of months whether it confers the chimeric Dectin-1 molecules the de novo to capacity to cross-present native OVA targeted to Dectin-1 via an antibody conjugate.

We thank the reviewer for suggesting this intriguing experiment. Our original conclusion was that the neck region of DNGR-1 was necessary for the proper function of the receptor. However, we did not experimentally address the question of whether it is on its own sufficient to confer the ability to promote cross-presentation to unrelated receptors. The suggested experiment has some caveats that are important to consider. MuTu DC1940 cells already express low levels of Dectin-1 endogenously although this should not be too much of a problem as overexpression of Dectin-1 in RAW cells, which also have low baseline levels of Dectin-1, has previously been shown to enhance Dectin-1 dependent responses (Gantner et al. *J. Exp. Med.*, 2003). More important is the fact that Dectin-1 is expressed as a monomer and its ligand-induced dimerisation has been suggested to be important for receptor function (Rogers et al. *Immunity*, 2005; Brown et al. *Protein Science*, 2007). Transplanting the neck region of DNGR-1 makes the receptor constitutively dimeric (see Fig 4B), a feature that could have unpredictable consequences. Nevertheless, we proceeded with the experiment as suggested. We used retroviral transduction of MuTu cells to overexpress WT Dectin-1 or a chimeric protein consisting of the neck region of DNGR-1 and the intracellular, transmembrane and ligand-binding domains of Dectin-1. As a control, we transduced MuTu cells with an empty-vector control virus. We FACS-sorted the transduced cells, so that expression levels of WT Dectin-1 and the chimeric protein were comparable, and we generated a conjugate of anti-Dectin-1 antibody (clone 2A11) covalently bound to an extended SIINFEKL peptide (following the approach detailed in Sancho et al, *JCI* 2008). Treatment of the control cells with this targeting reagent in a cross-presentation assay resulted in a baseline IFN- γ production by pre-activated OT-I T-cells, presumably due to the low levels of endogenously expressed Dectin-1. When we treated the cells transduced to overexpress WT Dectin-1, we observed approximately 10-fold shift in the dose response compared to the control cells, indicating that overexpression of Dectin-1 has measurable consequences in the assay. However, in the cells transduced with the chimeric protein, we observed no such gain-of-function suggesting that the chimera is inactive.

Antibody-mediated targeting may not allow receptor triggering in the same way as ligand binding. We therefore generated covalent complexes of laminarin, a Dectin-1 ligand, and ovalbumin (as per Xie et al. *Biochem. Biophys. Res. Commun.* 2010) and used those as an alternative method to targeting ovalbumin to MuTu cells. In line with what we observed using antibody targeting, the ovalbumin present in the laminarin complexes was cross-presented by control cells, as evidenced by production of IFN- γ by pre-activated OT-I T-cells. Overexpression of Dectin-1 caused an improvement in sensitivity as denoted by a shift in the dose-response curve. However, in the cells expressing the chimeric protein, no such improvement was apparent.

Taken together, these data suggest that the neck region of DNGR-1, while necessary for the function of DNGR-1 itself, is of its own not sufficient to confer the ability to promote cross-presentation to unrelated proteins. Importantly, as mentioned above, the data are difficult to fully interpret owing to the constitutively dimeric status of the chimeric protein which might have adverse effects on its ability to function. As such, we include the data here for the reviewer's appraisal, but we mention them in the manuscript only as data not shown (highlighted in yellow).

Figure 1. Following reduction the long isoform 'band' appears composed of two bands. This needs to be discussed. Based on Figure 1C, they do not appear to correspond to distinct glycoforms. We thank the reviewer for pointing out this omission. We think that the two bands do correspond to different glycoforms. The deglycosylation reaction with PNGase F (Fig 1C) that we performed will only remove N-bound glycans that do not contain an $\alpha(1-3)$ -Fucose bound to the core N-acetyl glucosamine. It is thus possible that the two bands correspond to glycoforms that differ in O-glycosylation or $\alpha(1-3)$ -fucosylated N-glycans. We have now clarified this in the text (highlighted in yellow).

How are the few multimers assembled (see also Figure 6D)?

The reduction-sensitive higher oligomers seen in the soluble ECD proteins (Fig 1), probably correspond to a fraction of the protein that failed to fold correctly and aggregated through the cysteines in the CTLD during protein production. Importantly, we could not observe any signs of increased reduction-insensitivity in these oligomers under any conditions, and they are effectively removed by size exclusion chromatography during protein purification. Crucially, they should not be confused with the higher, reduction-resistant oligomers that we observed in the context of the transmembrane protein (Fig 6D). As we speculated in the discussion, the formation of the reduction-resistant oligomers of full-length DNGR-1 could be mediated through the cysteines in the transmembrane or intracellular parts of the protein. We have now included further clarification in the discussion to prevent any confusion (highlighted in yellow).

Is there any additional Cys in the neck?

All murine isoforms contain only one cysteine residue in their neck region. The neck of the human DNGR-1 isoform has an additional cysteine, which is not conserved in other species. Whether this cysteine is also engaged in mediating dimerization of the receptor is currently unknown, but its presence could explain the higher proportion of the misfolded higher oligomers seen in human DNGR-1 ECD compared to the mouse isoforms (Fig 1A).

Figure 7. Is it necessary to use pre-activated OVA-specific CD8⁺ T-cells (OT-I) overnight?

We and others have invested significant effort into optimizing the in-vitro cross-presentation assay (Iborra et al. JCI, 2012, Hanc et al. Immunity 2015), and find that the sensitivity of pre-activated OT-I T cells increases the robustness of the cross-presentation assay with dead cell-associated antigens compared to using naïve OT-I. This might be due to the fact that IFN γ production by pre-activated OT-I cells is a direct readout of antigen presentation and not dependent on the activation status of the DCs. An additional advantage, pre-activation also results in T cell expansion, making OT-I from a single mouse sufficient for one experiment.

Accepted

15 September 2016

Thank you for submitting your revised manuscript to The EMBO Journal. Your study has now been re-reviewed by referee #2 and as you can see below the referee appreciates the introduced changes. I am therefore very pleased to accept the manuscript for publication here.

REFeree REPORT

Referee #2:

The authors have fairly addressed the points I raised and his rebuttal letter (to be published) is very informative.

Corresponding Author Name: Caetano Reis e Sousa

Manuscript Number: EMBOJ-2016-94695R